# Establishment and Comparison of Detection Methods for Ricin and Abrin Based on Their Depurination Activities

**DOI:** 10.3390/toxins17040177

**Published:** 2025-04-03

**Authors:** Lina Dong, Tingting Liu, Jiaxin Li, Cen Wang, Jing Lv, Jing Wang, Jinglin Wang, Shan Gao, Lin Kang, Wenwen Xin

**Affiliations:** 1School of Basic Medicine, Anhui Medical University, Hefei 230032, China; donglina2024@163.com; 2State Key Laboratory of Pathogen and Biosecurity, Academy of Military Medical Sciences, Beijing 100071, China; liutingtingwushi@163.com (T.L.); ljx1658675586@163.com (J.L.); jinglv1205@163.com (J.L.); amms_wj@163.com (J.W.); wilwj10801@sina.com (J.W.); gaoshan845@163.com (S.G.); 3Department of Public Health, School of Public Health and Health Management, Gannan Medical University, Ganzhou City 341000, China; cenwang1003@163.com

**Keywords:** abrin, ricin, N-glucosidase activity, detection, HPLC–MS/MS

## Abstract

Ricin (RT) and abrin (AT) are plant toxins extracted from *Ricinus communis* and *Abrus precatorius*, respectively, and both have N-glycosidase activity. The detection of these toxins is vital because of their accessibility and bioterrorism potential. While ricin can be effectively detected based on its depurination activity, only a few tests are available for detecting the depurination activity of abrin. Therefore, it is unclear whether they share the same optimal reaction substrate and conditions. Here, we established optimum depurination conditions for ricin and abrin, facilitating the in vitro detection of their depurination activity using high-performance liquid chromatography–tandem mass spectrometry. The parameters optimized were the reaction substrate, bovine serum albumin (BSA), buffer, pH, temperature, time, antibodies, and magnetic beads. Both toxins showed better depurination with single-stranded DNA. However, substrate length, adenine content, BSA concentration, buffer concentration, reaction temperature, and reaction time differed between the two toxins. The optimal conditions for ricin depurination involved a reaction in 1 mM ammonium acetate solution (0.5 μM DNA15A, 20 μg/mL BSA, and 1 mM Zn^2+^, with pH 4.0) at 55 °C for 1 h. The optimal conditions for abrin depurination involved a reaction in 1 mM ammonium citrate solution (0.2 μM DNA20A, 10 μg/mL BSA, 1 mM Mg^2+^, and 0.5 mM EDTA, with pH 4.0) at 45 °C for 2 h. After optimization, the limits of detection (LOD) for ricin and abrin were 0.506 ng/mL and 0.168 ng/mL, respectively. The detection time was also significantly reduced.

## 1. Introduction

Ricin (RT) and abrin (AT) are highly potent plant toxins derived from *Ricinus communis* and *Abrus precatorius*, respectively. Both toxins are type II ribosome-inactivating proteins and have similar structure and function. Both are heterodimeric and each consists of two distinct polypeptide chains: A chain and B chain [1,2]. Their A chains exhibit N-glycosidase activity. Owing to this enzymatic function, A chains can catalyze the hydrolysis of the N-glycosidic bond of adenine residue within large ribosomal RNA (rRNA), leading to adenine detachment. The B chains are crucial to cellular uptake. They contain multiple lectin-like domains that bind to specific cell-surface glycoconjugates. This binding enables the B chains to mediate endocytosis of the entire toxin complex into target cells. Once internalized, the A chain is translocated into the cytosol, where it exerts its toxic effect by inhibiting protein synthesis, ultimately leading to cell death. Both ricin and abrin have a molecular weight of approximately 65 kDa [3,4]. The homology of both A chains is approximately 40%. The abrin B chain has 60% homology with the ricin B chain [5,6]. The median oral lethal doses (LD_50_) of ricin and abrin for humans are 0.1–1 μg/kg and 0.01–0.04 μg/kg, respectively [7,8]. The structural similarities, high toxicity, and readily accessible sources of these two toxins make them relatively easy to produce. They are considered potential agents for biological warfare and bioterrorism and significant threats to public health and safety. Ricin has been used in assassinations and intimidation. Between 2010 and 2022, ricin was reported in about 10 terrorist attacks. Among them was a letter to President Donald Trump containing ricin powder [9]. Abrin, because of its bright color, is often sold as a handicraft—someone once pricked their finger and died of poisoning when making an abrin bracelet—and recently there have been cases of poisoning caused by eating abrin beans by mistake [10,11]. Consequently, the development of rapid and accurate detection methods for ricin and abrin is critical.

The detection methods for abrin and ricin can be classified into three distinct categories. The first category is immunological detection methods that use specific antibodies. These methods include enzyme-linked immunosorbent assay (ELISA) [12], Western blotting, immunochromatography assay (ICA), colloidal gold immunochromatography strip (CGIS) [13], and electrochemiluminescence (ECL) [14,15]. The advantages of these immunological detection methods include high sensitivity and straightforward operation. However, their performance tends to be contingent on the specificity of the antibodies employed. These methods place stringent demands on the quality and specificity of the antibodies, as the reliability of the detection results is highly correlated with the ability of the antibodies to selectively bind to the target toxins. The second category of detection methods is those that use mass spectrometry-based analysis and identification techniques. These methods can precisely discern the amino-acid sequence of toxin-specific peptides. Moreover, these methods enable the identification and typing of toxins with a high degree of accuracy, but the experimental process is lengthy and complicated [16,17,18]. The third category of detection method focuses on detecting depurination activity and cytotoxic activity [17,19,20,21,22]. The activity-based detection method capitalizes on the N-glycosidase activity inherent in ricin and abrin. These toxins act upon a substrate and catalyze a depurination reaction, thereby exerting their enzymatic function. The amount of depurination induced by the toxins can be quantitatively measured, enabling the determination of toxin levels.

Current ricin and abrin detection technologies include immunoassays, nucleic acid detection systems, electrochemical sensors, and mass spectrometry. Most methods cannot determine toxin activity, and only active toxins pose a threat to humans. Thus, establishing a detection method that can directly reflect the biological activity of toxins is crucial. Extant research has shown that ricin activity can be effectively detected through quantification of the substrate depurination it induces [23]. However, there are only a few tests for the depurination activity of abrin. Therefore, it is currently unknown whether the optimal reaction substrate and reaction conditions of the two toxins are the same. To address this gap in the literature, we established and optimized a high-performance liquid chromatography–tandem mass spectrometry (HPLC–MS/MS) method of adenine detection and optimized the components and reaction conditions of ricin and abrin reaction systems, including the reaction substrate and its concentration, bovine serum albumin (BSA) concentration, reaction buffer concentration, pH, reaction temperature, reaction time, magnetic beads, and capture antibody. Finally, the established detection method was evaluated (Figure 1).

## 2. Results and Discussion

### 2.1. Evaluation of the Synthetic DNA and RNA Substrates of Ricin and Abrin

To determine whether ricin and abrin have different substrate preferences and identify their respective optimal substrates, we tested several synthetic DNA and RNA oligomers as potential depurination substrates. These included single-stranded DNA, single-stranded RNA, double-stranded DNA, and neck-ring DNA and neck-ring RNA structures. The single-stranded substrates included straight-strand and stem-ring structures. The single-stranded structure contained adenine of different lengths. It is known that thymine can promote the shedding of adenine in double-stranded DNA [24]. Therefore, we investigated the effects of thymine on the depurination activity of the toxins. One is to ensure that the length of the substrate chain is unchanged and the thymine content is increased. The other is that the substrate contains only thymine and adenine, and as the thymine and adenine content increases, the length of the chain increases. Moreover, we designed a stem-ring-structure substrate to mimic the action of ricin and abrin on the ring-structure fragment of the natural ribosome rRNA 28S subunit [21]. With the increase in the GA structure, the ring size and neck length of the substrate also gradually increased. This effect was used to investigate the effect of the stem-ring size on the activity of the substrate enzymes.

We found that for ricin, DNA15A (AAAAAAAAAAAAAAA) had the highest depurination ratio. As the amount of adenine increased, the amount of depurination on the single-stranded DNA decreased (Figure 2a). Single-stranded RNA and double-stranded DNA showed poor overall depurination effects and relatively higher negative ratios; therefore, they were not suitable as substrates (Figure 2b,c). The addition of thymine promoted the shedding of adenine in the stem-ring DNA structure; however, as the amount of thymine increased, the amount of depurination decreased. Increases in the GA content had little effect on the amount of depurination (Figure 2d). For the stem-ring RNA structure, RNA14-2GA (CGCGCGAGAGCGCG) without an uracil base had the best depurination effect (Figure 2e). Therefore, we speculated that uracil might inhibit the depurination effect of toxins. Finally, using the ratio method, we found that DNA15A was the best substrate with the highest ratio. Further optimization of the substrate concentration of DNA15A was made on the basis that the ratio was highest when the concentration of DNA15A was 0.5 μM (Figure 2f).

For abrin, the depurination activity in most single-stranded DNA and single-stranded RNA increased with increased amounts of adenine (Figure 2g,h). The depurination effect of the double-stranded DNA was poor, and increased thymine content had very little effect on the depurination activity (Figure 2i). The depurination of the stem-ring DNA structure was consistent with the depurination of ricin. The addition of thymine promoted the depurination reaction of the abrin; however, it was not enhanced by increasing thymine levels (Figure 2j). The stem-ring RNA structure showed no obvious pattern, and the presence of uracil did not alter the effect of the toxin on the substrate (Figure 2k). Finally, DNA20T-2GA has the highest ratio, but considering that DNA20T-2GA is not much higher than DNA20A (AAAAAAAAAAAAAAAAAAAA), and the DNA20A substrate is easier to make, DNA20A was selected as the substrate for subsequent experiments (Figure 2g–k). A concentration of 0.2 μM was selected for the abrin substrate DNA20A, as this showed the highest ratio and best depurination detection (Figure 2l).

### 2.2. Optimization of Conditions for Ricin and Abrin Depurination Reactions

To accurately establish the optimal reaction conditions, a systematic optimization of the reaction systems was performed. This encompassed a detailed optimization of multiple crucial factors, including the pH of the reaction mixture, contents and their concentrations, reaction temperature, and reaction time. In the context of the depurination reaction, BSA is known to confer a certain level of protection to the toxin. This allows the toxin to maintain its structural integrity and enzymatic activity, enabling it to exert its depurination function more effectively. Notably, the degree of protection provided by BSA varies depending on its concentration. Different BSA concentrations can lead to distinct interactions with the toxin. This impacts the overall efficiency and outcomes of the depurination reaction [15,25]. We found that the optimal concentrations of BSA for ricin and abrin were 20 μg/mL and 10 μg/mL, respectively (Figure 3a).

Furthermore, the activity of ricin and abrin in ammonium formate, ammonium acetate, and ammonium citrate solutions was investigated. We found that both toxins exhibited better depurination effects in the ammonium acetate buffer liquid system than in other solutions (Figure 3b,c). A previous study has shown that adding ethylene diamine tetraacetic acid (EDTA) and metal ions to the reaction system can improve toxin activity [26]. Therefore, Zn^2+^, Mg^2+^, Ca^2+^, and EDTA were added to the ammonium acetate buffer (Figure 3d). For ricin, the addition of Zn^2+^ and Ca^2+^ to the system enhanced its activity, whereas Mg^2+^ had no significant effect. The activity of ricin was also improved with the addition of EDTA. Therefore, 1 mM ammonium acetate containing 1 mM Zn^2+^ was selected as the buffer for ricin. For abrin, the depurination activity of the toxin was not improved by Zn^2+^ or Ca^2+^. The depurination activity of the toxin only significantly increased when Mg^2+^ and EDTA were added to the ammonium citrate buffer at the same time. Therefore, 1 mM ammonium citrate (containing 1 mM Mg^2+^ and 0.5 mM EDTA) was selected as the buffer for the final abrin system (Figure 3d). These optimal buffers of ricin and abrin were used to adjust the pH. The optimal pH value of ricin and abrin was approximately 4.0, which is consistent with previous research [27]. We also found that the pH tolerance range of the two toxins was extremely narrow and the tolerance window was very small. For ricin, when pH was 3.8–4.2, the toxin activity first increased and then decreased. Depurination activity was the highest when the pH was 4. However, when the pH exceeded 4.6, the depurination activity of ricin almost disappeared (Figure 3e). Abrin was marginally less affected by pH. It showed stable toxin activity when the pH was between 3.85 and 4.45, and when the pH was >5.75, the depurination activity of abrin almost disappeared (Figure 3f).

The depurination reactions of the toxins were significantly influenced by temperature variations. A series of experiments was performed by establishing a reaction temperature gradient. We found that as the reaction temperature increased, the extent of depurination in both toxins initially increased and then decreased. For ricin, the amount of adenine from the substrate reached its peak at 55 °C. Therefore, this was determined to be the optimal temperature for ricin-mediated depurination. For abrin, the optimal temperature for its depurination reaction was 45 °C. These findings underscore the critical role of temperature in modulating the enzymatic activities of ricin and abrin during depurination (Figure 3g).

A depurination test was performed by establishing a reaction-time gradient for the toxins, ranging from 0 to 24 h. The results demonstrated that as the reaction time increased, the degree of depurination by the toxins gradually increased. For ricin, the increase was relatively slow and tended to stabilize after 6 h of reaction. For abrin, the amount of depurination continued to increase gradually even after 24 h of reaction. Concurrently, the amount of depurination of the reaction substrates for both toxins was also augmented. Based on the ratio method, which considers the relationship between the degree of depurination and other relevant factors such as background noise and experimental efficiency, the optimal reaction times for ricin and abrin were determined to be 1 h and 2 h, respectively (Figure 3h,i).

After optimization, the optimum conditions for ricin depurination were as follows: reaction in 1 mM ammonium acetate solution (containing 0.5 μM DNA15A, 20 μg/mL BSA, and 1 mM Zn^2+^, with a pH of 4.0) at 55 °C for 1 h. The optimal reaction conditions for abrin were reaction in 1 mM ammonium citrate (containing 0.2 μM DNA20A, 10 μg/mL BSA, 1 mM Mg^2+^, 0.5 mM EDTA, with a pH of 4.0) at 45 °C for 2 h.

### 2.3. Selection of Trapping Antibodies and Magnetic Beads

Trapping antibodies play an important role in the detection of toxins in complex substrates and improvement of detection sensitivity [28,29]. Therefore, we compared the effects of multiple antibodies and magnetic beads to improve the detection of toxins. We selected nanobodies that could only bind to the B chain of each toxin. This characteristic ensured that the nanobodies would not interfere with the enzymatic activity of the A chain, leading to more accurate and reliable detection results. We found the poorest effect with pAb-RT and the best effect with the B4 nanobody (Figure 4a). However, the capture effect of pAb-AT was much higher than that of the nanobody (Figure 4b). Therefore, the best trapping antibodies for ricin and abrin were B4 and pAb-AT, respectively. We performed antibody affinity tests for the ricin and abrin antibodies B4 and pAb-AT. The affinity between ricin and the B4 antibody was 8.39 × 10^−9^ M, R^2^ = 0.99. The affinity between abrin and pAb-AT was 1.38 × 10^−11^ M, R^2^ = 0.99. An evaluative diagram of ricin and abrin antibody affinities is provided in the Appendix A (Appendix A). A comparative analysis of multiple magnetic-bead variants revealed significantly superior capture efficiency in Dynabeads™ M-270 streptavidin compared with other magnetic beads. Therefore, Dynabeads™ M-270 streptavidin was used in the subsequent experiments (Figure 4c,d).

### 2.4. Linearity Range and Sensitivity of the Detection Method

Ricin and abrin in water samples were captured using immunomagnetic beads and then added to the reaction mixture, including an internal standard for HPLC–MS/MS analysis. The addition of the internal standard was intended to minimize the variability in the results, thereby enhancing the accuracy and reliability of our quantitative analysis [30]. Ricin and abrin concentrations were logarithmically transformed for the horizontal coordinate, whereas the ratio of toxin-induced substrate depurination to substrate self-depurination served as the ordinate. A nonlinear fitting curve was established for quantitative toxin detection. Ratios > 5 indicated positive detections. Ricin without magnetic-bead enrichment had an LOD of 25.98 ng/mL, showing a good linear relationship in the range of 100–1600 ng/mL (Figure 5a,b). With immunomagnetic beads for toxin enrichment, the LOD for abrin was 0.168 ng/mL, showing a good linear relationship in the range of 0.781–50 ng/mL (Figure 5c,d). Ricin without magnetic-bead enrichment was detected at an LOD of 37.63 ng/mL, showing a good linear relationship in the range of 62.5–2000 ng/mL (Figure 5e,f). With immunomagnetic beads for toxin enrichment, ricin was detected at 0.506 ng/mL, showing a good linear relationship in the range of 0.391–100 ng/mL (Figure 5g,h). After the addition of the magnetic beads, the sensitivity of abrin and ricin increased by 154 times and 74 times, respectively.

### 2.5. Detection of Simulated Samples

We prepared samples of ricin and abrin in 1× phosphate-buffer saline (PBS), milk, and human serum. The nonlinear and linear fitting curves are shown in Figure 6. The detection limits, detection ranges, and linearity of ricin and abrin in different samples are shown in Table 1. The LOD of abrin in PBS was 0.579 ng/mL, showing a good linear relationship in the range of 6.25–400 ng/mL. The LOD of abrin in milk was 1.204 ng/mL, showing a good linear relationship in the range of 6.25–400 ng/mL. The LOD of abrin in serum was 1.458 ng/mL, showing a good linear relationship in the range of 3.125–200 ng/mL. The LOD of ricin in PBS was 1.717 ng/mL, showing a good linear relationship in the range of 3.125–200 ng/mL. The LOD of ricin in milk was 3.984 ng/mL, showing a good linear relationship in the range of 12.5–200 ng/mL. The LOD of ricin in serum was 3.978 ng/mL, showing a good linear relationship in the range of 6.25–100 ng/mL.

Simulated sample evaluations showed that the sensitivity of the PBS samples of ricin and abrin was 2–3-fold lower than that of water samples. PBS, a commonly used buffer, contains salt ions and has a pH of approximately 7.4, whereas the optimal reaction pH of ricin and abrin in this study was around 4. In addition, salt ions in PBS solution can affect mass spectrometry-based detection and nanobody toxin capture, lowering PBS sample sensitivity. Milk and serum samples showed a greater sensitivity loss than water samples. Milk contains proteins and fats: proteins noncompetitively bind to toxin active sites, inhibiting toxin activity, while fats interfere with mass spectrometry signals. Serum contains proteins, which may nonspecifically adsorb antibodies, reducing toxin capture efficiency. High serum salt concentrations markedly reduce mass spectrometry sensitivity, and serum is also weakly alkaline. Furthermore, analysis of the simulated samples often necessitated additional pretreatments of the samples. These procedures (such as filtration, extraction, and purification) could have resulted in some loss of the toxin content in the samples. Consequently, the amount of toxin available for detection was further reduced, contributing to the decline in detection sensitivity.

### 2.6. Recovery Rates of the Detection Method

The immunocapture recovery rates of the toxins in these distinct matrices were systematically evaluated. The results are presented in Table 2. The recovery rates of active ricin ranged from 82.6% to 121.8%, whereas those of active abrin ranged from 93.9% to 124.1%. These findings demonstrate that the proposed method was capable of effectively recovering the target substances within an acceptable error margin. Notably, in the milk and serum samples, the recovery rates exceeded 120% for some samples. This phenomenon could potentially be attributed to the relatively high level of interference in milk and serum matrices, which may have affected the quantification process and led to deviations in the measured recovery rates.

### 2.7. Specificity of the Detection Method

Toxins from different origins, including *clostridium perfringens* epsilon toxin (ETX), diphtheria toxin (DT), staphylococcal enterotoxin A (SEA), botulinum toxin A (BoNT A), ricin, and abrin, were used to evaluate the specificity of our assay. We found a small degree of cross-reaction between ricin and abrin, but no cross-reactions between other toxins (Figure 7a,b). As abrin concentration increased, a minor cross-reaction with ricin occurred. At an abrin concentration of 10 μg/mL, the detection ratio was 8.7. When this ratio was substituted into the ricin quantification equation, it corresponded to a calculated ricin concentration of 0.49 ng/mL, yielding a cross-reaction rate of 0.0049% (Figure 7c). With 100 ng/mL of ricin, the ratio was 3.5. As ratios < 5 were considered negative, no cross-reaction was found to occur below 100 ng/mL (Figure 7d). These results show that the specificity of the ricin and abrin assay was good.

## 3. Conclusions

We optimized the conditions for ricin and abrin detection using HPLC–MS/MS based on their N-glycosidase activity, and the similarities and differences between ricin and abrin were compared. The detection method we established for abrin and ricin is sensitive, fast, and requires only a small sample volume. It is also able to determine whether the toxins in a sample are active. We found that both ricin and abrin had a preference for single-stranded DNA. Despite their similar structures and functions, the two toxins had different optimal reaction substrates, reaction buffers, incubation temperatures, and incubation times. The optimal conditions for ricin were reactions in 1 mM ammonium acetate solution (containing 0.5 μM DNA15A, 20 μg/mL BSA, and 1 mM Zn^2+^, with a pH of 4.0) at 55 °C for 1 h, whereas those for abrin were reactions in 1 mM ammonium citrate (containing 0.2 μM DNA20A, 10 μg/mL BSA, 1 mM Mg^2+^, and 0.5 mM EDTA, with a pH of 4.0) at 45 °C for 2 h. Ultimately, the optimized LODs for ricin and abrin were 0.506 ng/mL and 0.168 ng/mL, respectively. The results of this study provide a reference point for further studies on the detection of these two toxins.

## 4. Materials and Methods

### 4.1. Materials

Dynabeads^®^ Antibody Coupling Kit, Dynabeads^®^ M-450 epoxy, Dynabeads™ M-280 streptavidin, Dynabeads™ M-270 streptavidin, Dynabeads™ MyOne™ T1 streptavidin, acetonitrile (HPLC grade), and methanol (HPLC grade) (>98%) were purchased from Thermo Fisher Scientific (Lake Success, NY, USA). In addition, 0.5 M EDTA (pH 8.0) was purchased from Solarbio Science Technology (Beijing, China). All substrates and ammonia were synthesized by Genscript Biotech Corporation (Nanjing, China). BSA, ammonium citrate, ammonium formate, and ammonium acetate were purchased from Sigma-Aldrich (St. Louis, MO, USA). Ultrapure water was purchased from Wahaha Pure Water Company (Hangzhou, China). Adenine and aciclovir were purchased from Merck (Darmstadt, Germany). Abrin, ricin, and abrin polyclonal antibodies (pAb-AT) were prepared as previously described [13,31]. The ricin polyclonal antibody (pAb-RT) was purchased from the Abcam Company. PBS was obtained from Coolaber Technology (Beijing, China). Serum was provided by our local hospital. Milk was bought from the local store.

### 4.2. Designs and Sequences of the Substrates

We designed various types of oligonucleotide chain substrates, including both DNA and RNA featuring straight-chain or stem-ring structures, with varying lengths and adenine content. The substrates were synthesized by Genscript Biotech Corporation (Nanjing, China). Detailed sequence information of the substrates used in this study is presented in the Appendix A (Appendix A).

### 4.3. Nanobody Design and Purification

Nanobodies of ricin and abrin were prepared. The sequences of the anti-ricin nanobodies B4, 21-B4, G5/B7, and G5/B9 and of the anti-abrin nanobodies Abr3E, Abr10C, Abr5, and Abr11 were obtained from the literature. These antibodies were constructed on the pET-28a (+) plasmid with *Nco*I and *Xho*I cleavage sites, expressed in *E. coli* BL21 (DE3) with a solubility-enhancing Trx tag, and purified by adding a 6×his tag [32,33,34,35,36]. Protein-expressing strains were cultured at 180 rpm and 30 °C. When OD reached 0.6, IPTG was added to a final concentration of 0.4 mM, and expression was induced at 16 °C overnight. Proteins were ultrasonically lysed and then purified via affinity chromatography (HisTrap^TM^ HP, Ct. Fairfield, GE Healthcare, Chicago, IL, USA) and gel chromatography (HiPrep SepHacry1^TM^ S-100 HR, Ct. Fairfield, GE Healthcare, USA) to obtain a high-purity nanobodies. The nanobody sequences are shown in Appendix A.

### 4.4. HPLC–MS/MS Analysis

HPLC–MS/MS was conducted using the ACQUITY UPLC^®^I-Class-Xevo TQ-S mass spectrometry device (Waters Corporation, Milford, MA, USA). Adenine was separated via C18 reversed-phase chromatography (300 Å, 1.7 μm, 2.1 mm × 150 mm) (ACQUITY UPLC@BEH C18). The column temperature was maintained at 4 °C. The mobile phases comprised 10 mM ammonium formate solution (pH 4.0) for the aqueous phase and 100% methanol for the organic phase. The flow rate was 0.2 mL/min. The gradient elution program was 0–5 min (10–90% B), 5–6 min (90% B), 6–8 min (90–10% B), and 8–10 min (10% B). The injection volume of the sample was 2 μL. The mass spectrometry utilized UniSpray+ as the ion source. Data acquisition was conducted in the positive ion mode through multireaction monitoring (MRM). The desolvation temperature was set at 500 °C, and the source temperature was 150 °C. Qualitative and quantitative analyses were performed using the ratio of the peak area of depurination induced by the toxin to the baseline depurination content of the substrate. Ratios > 5 were considered positive. The MRM ion pairs, fragmentation voltages, and collision energies of the adenine and internal standard acyclovir target objects are shown in Table 3. Experiments showed retention times of 2.55 and 2.47 min for adenine and internal standard acyclovir, respectively. These close but nonoverlapping retention times confirmed that acyclovir addition did not affect toxin detection.

### 4.5. Optimization of Reaction Conditions

The substrates of ricin and abrin were screened separately. Ricin and abrin act on different types of single-stranded DNA, single-stranded RNA, and double-stranded DNA. The optimal reaction concentration of each substrate was also optimized. The final concentration of both substrates was set to 0.02–2 μM. The optimal reaction concentration of BSA was explored and set to 0–100 μg/mL. The reaction effects of different concentrations of ammonium formate buffer (0–20 mM), ammonium acetate buffer (0–20 mM), ammonium citrate buffer (0–20 mM), and their buffers with metal ions and EDTA were compared. After determining the optimal reaction buffers and their reaction concentrations, the optimal buffer pH was further optimized to a pH range of 3.25–5.75. The optimal reaction temperatures were investigated and set as 4–66 °C. The optimal reaction times were investigated and set as 0–24 h. The antibodies of ricin and abrin were compared and selected. Antibody affinity for the screened antibodies B4 and pAb-Ab was further evaluated. In the experiment, 0.02% PBST buffer was consistently used, and an SA sensor was used for affinity detection between ricin and B4. Ricin was diluted to 0–20 nM with 0.02% PBST, and the curing concentration of biotinylated B4 antibody was 10 μg/mL. A ProA sensor was used for affinity detection between abrin and pAb-AT, abrin was diluted with 0.02% PBST to 0–40 nM, and the curing concentration of pAb-AT antibody was 5 μg/mL. To better enrich the toxins, different types of immunomagnetic beads were tested, and the most suitable were selected.

### 4.6. Linearity Range and Sensitivity

In the absence of magnetic-bead enrichment, the sensitivity of the method was directly explored using the optimized reaction system and conditions. Using the ratio dilution method with the optimized reaction system, abrin was diluted with sterile water to a concentration range of 0.781–3200 ng/mL for detection, whereas ricin was diluted to a concentration range of 7.813–12,000 ng/mL for detection. Sterile water was used as a negative control. The sensitivity of the method was investigated after enrichment of the toxins by immunomagnetic beads, and quantitative standard curves were constructed. To mitigate any experimental errors arising from the procedures, the internal standard acyclovir was incorporated during the sample detection process. For magnetic-bead enrichment detection, abrin was diluted with sterile water to a concentration range of 0.195–200 ng/mL and ricin to a concentration range of 0.098–100 ng/mL. Abrin and ricin were incubated with their respective antibody beads at room temperature for 1 h, and the mixtures were adsorbed by a magnetic rack for 2–3 min. The supernatants were then removed and the beads were added to the reaction system. Immediately after the reaction was completed, ammonia water was added to terminate the reaction, and the adenine content was measured through HPLC–MS/MS.

### 4.7. Complex Matrices and Recovery Rate Detection

The sensitivity and recovery of the method were evaluated in milk, 1× PBS, and human serum substrates. Using the ratio dilution method, the detected concentrations of ricin and abrin were found to range from 0.098 to 400 ng/mL. Sterile water was used as a negative control. The magnetic beads of the conjugated antibodies were gently blown with the toxins prepared using 1× PBS, human serum, and milk and incubated at room temperature for 1 h. After the magnetic beads had been adsorbed by the magnetic rack and the supernatant removed, the beads were cleaned with sterile water three times before adding them to the reaction system. The rest of the reaction steps were same as described in Section 4.6 (unless otherwise stated). To determine the recovery rate, ricin and abrin were diluted to 50 ng/mL, 100 ng/mL, and 200 ng/mL with PBS, serum, and milk and added into the reaction system. Finally, the ratio was substituted into the corresponding complex matrix standard curve to calculate the toxin concentrations and recovery rates.

### 4.8. Evaluation of the Specificity of the Detection Method

To comprehensively determine the specificity of the ricin and abrin detection method, *clostridium perfringens* epsilon toxin, diphtheria toxin, staphylococcal enterotoxin A, botulinum toxin A, abrin, and ricin were individually diluted to a concentration of 100 ng/mL. Considering the degree of similarity in the structural sequences of ricin and abrin, there was potential for cross-reactions to occur. Thus, a higher toxin concentration gradient was established to explore the cross-reactivity between ricin and abrin. In particular, ricin and abrin were captured with antibody B4, and the toxin concentrations were set to 10 μg/mL, 1 μg/mL, and 0.1 μg/mL. Ricin and abrin were captured with the antibody pAb-AT and the toxin concentrations were set to 100 ng/mL, 10 ng/mL, and 1 ng/mL. Following the incubation period, HPLC–MS/MS was used to determine adenine concentrations.

## Figures and Tables

**Figure 1 toxins-17-00177-f001:**
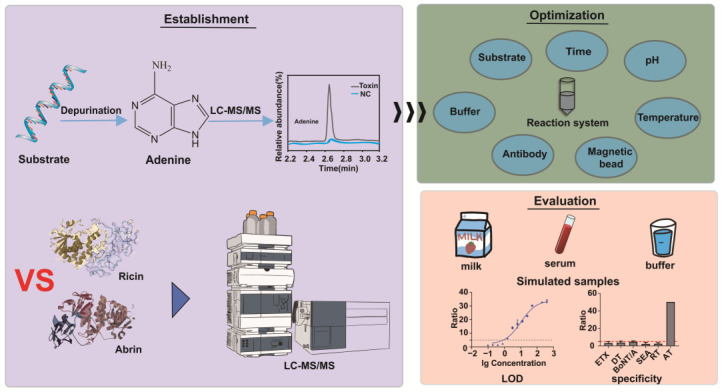
An illustration of the overall research strategy in this study.

**Figure 2 toxins-17-00177-f002:**
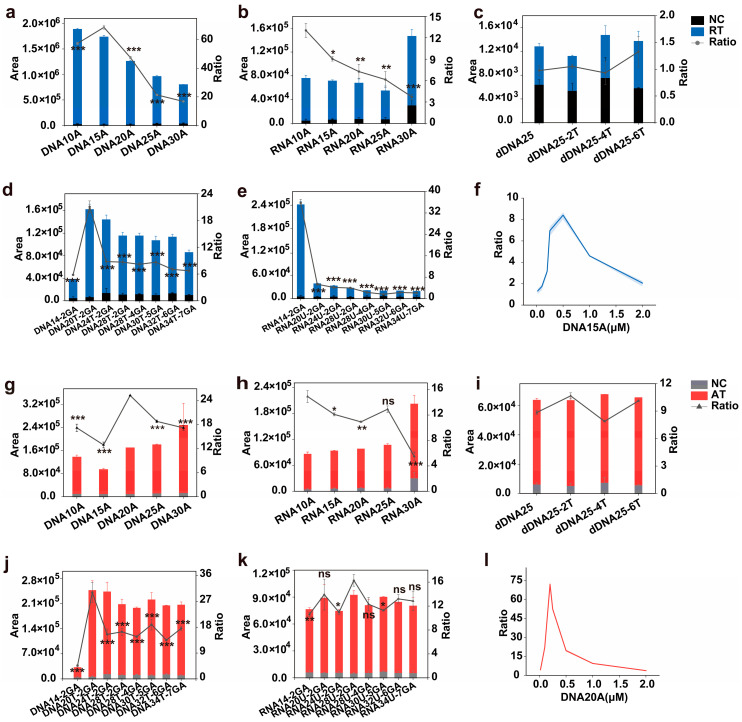
Ricin and abrin reaction substrates and their optimal concentrations. (**a**–**e**) Ricin (5 μg/mL) reacts with single-stranded DNA, single-stranded RNA, double-stranded DNA, neck-ring DNA, and neck-ring RNA structures. (**f**) Ricin (0.2 μg/mL) reacts with varying DNA15A concentrations. (**g**–**k**) Abrin (5 μg/mL) reacts with single-stranded DNA, single-stranded RNA, double-stranded DNA, neck-ring DNA, and neck-ring RNA structures. (**l**) Abrin (0.02 μg/mL) reacts with varying DNA20A concentrations. Peak adenine detection levels are shown. They y-axis represents the ratio of substrate depurination due to toxin activity versus substrate self-depurination. Sterile water replaced the toxin as a negative control, with all other reaction components unchanged. The amount of adenine shed by the substrate itself was assessed. ns: *p* > 0.05; *: *p* < 0.05; **: *p* < 0.01; ***: *p* < 0.001.

**Figure 3 toxins-17-00177-f003:**
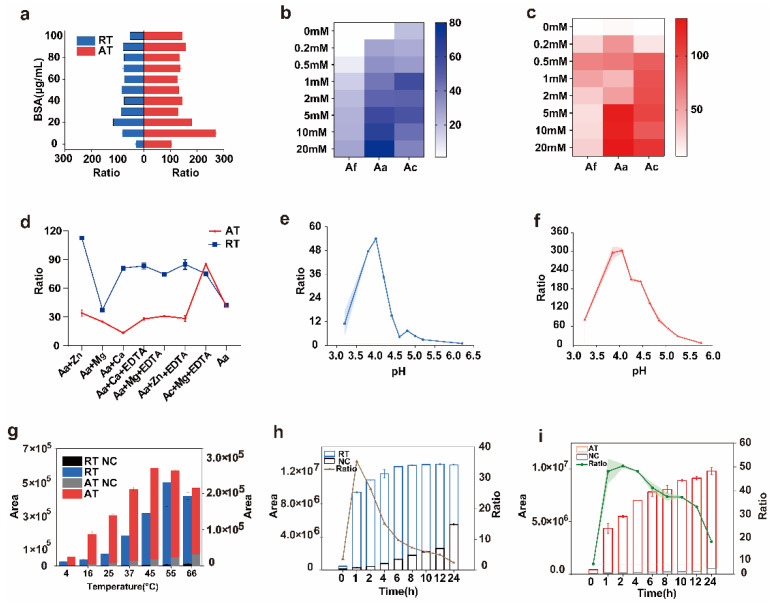
The optimization of the reaction conditions for ricin and abrin. (**a**) The depurination of ricin and abrin in different concentrations of BSA. The concentrations of ricin and abrin were 2 μg/mL and 0.4 μg/mL; (**b**) the depurination of ricin with different buffer solution concentrations. The concentration of ricin was 5 μg/mL; (**c**) the depurination of abrin with different concentrations of the buffer solution. The concentration of abrin was 5 μg/mL; (**d**) the depurination of ricin and abrin in buffered solutions containing Zn^2+^, Mg^2+^, Ca^2+^, or EDTA. The concentration of ricin and abrin was 5 μg/mL; (**e**) ricin depurination activity at different pH levels. The concentration of ricin was 2 μg/mL; (**f**) abrin depurination activity at different pH levels. The concentration of ricin was 0.4 μg/mL; (**g**) the depurination of ricin and abrin at different temperatures. The concentrations of ricin and abrin were 10 μg/mL and 5 μg/mL, respectively; (**h**) ricin depurination with different incubation times. The concentration of ricin was 0.2 μg/mL; (**i**) abrin depurination with different incubation times. The concentration of abrin was 0.04 μg/mL. Aa, ammonium acetate; Aa + EDTA, ammonium acetate + 5 mM EDTA; Ac, ammonium citrate; Ac + EDTA, ammonium citrate + 5 mM EDTA; Af, ammonium formate; BSA, bovine serum albumin; EDTA, ethylene diamine tetraacetic acid.

**Figure 4 toxins-17-00177-f004:**
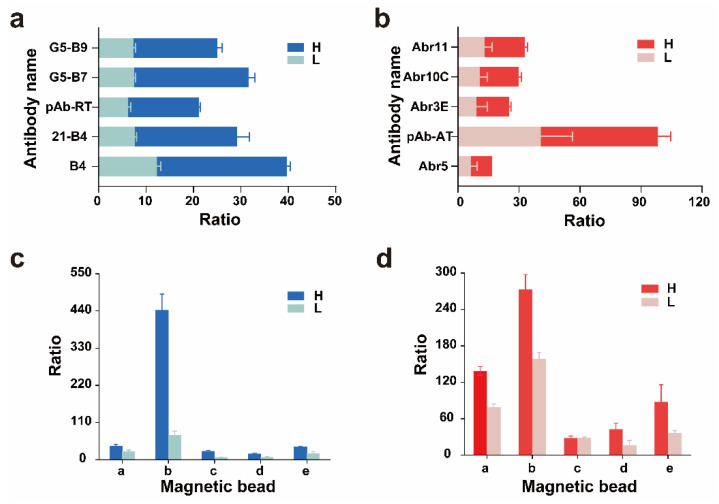
Effects of different magnetic beads and trapping antibodies on ricin and abrin depurination. (**a**) Five ricin antibodies were coated with magnetic beads and depurination was performed; (**b**) five abrin antibodies were coated with magnetic beads and depurination was performed; (**c**) five types of magnetic beads were coated with the best antibodies of ricin and depurination was performed; (**d**) five types of magnetic beads were coated with the best antibodies of abrin and depurination was performed. The magnetic-bead amount was 30 μL (1 mg/mL), and the amount of antibody used was 30 μg (water matrix). The volume of each sample was 500 μL. The high (H) and low (L) concentrations of ricin and abrin were 250 and 50 ng/mL, respectively. a: Dynabeads^®^ Antibody Coupling Kit; b: Dynabeads™ M-270 Streptavidin; c: Dynabeads™ M-280 Streptavidin; d: Dynabeads^®^ M-450 epoxy; e: Dynabeads™ MyOne™ T1 Streptavidin.

**Figure 5 toxins-17-00177-f005:**
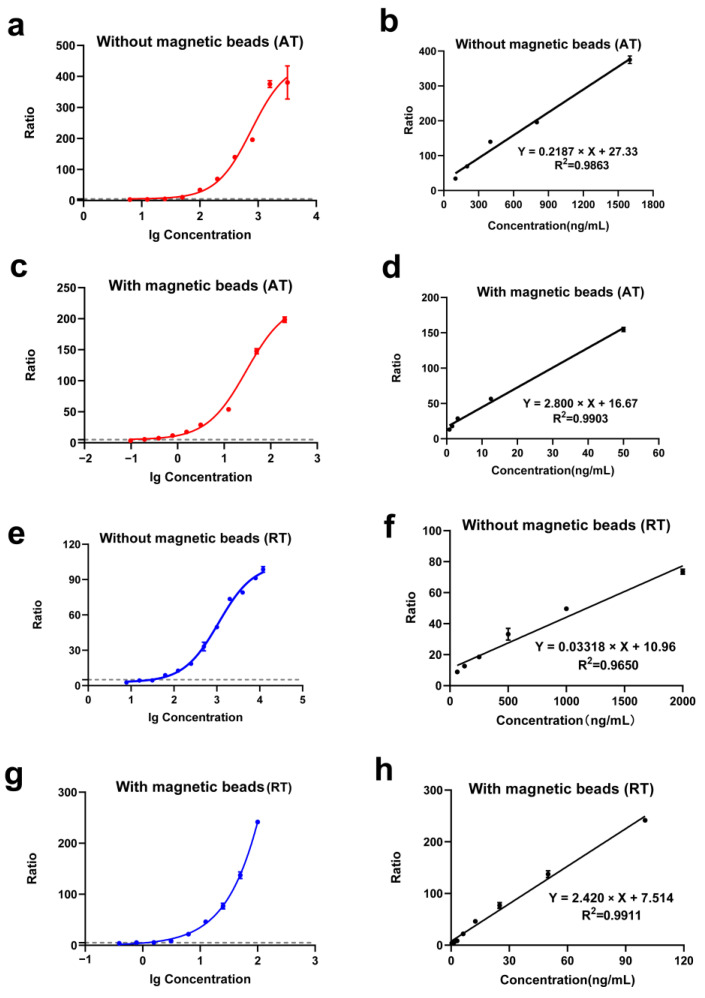
Evaluation of abrin and ricin detection methods. (**a**) Nonlinear fitting curve for abrin without immunomagnetic-bead enrichment; (**b**) linear fitting curve for abrin without immunomagnetic-bead enrichment; (**c**) nonlinear fitting curve for abrin with immunomagnetic-bead enrichment; (**d**) linear fitting curve for abrin with immunomagnetic-bead enrichment; (**e**) nonlinear fitting curve for ricin without immunomagnetic-bead enrichment; (**f**) linear fitting curve for ricin without immunomagnetic-bead enrichment; (**g**) nonlinear fitting curve for ricin with immunomagnetic-bead enrichment; (**h**) linear fitting curve for ricin with immunomagnetic-bead enrichment.

**Figure 6 toxins-17-00177-f006:**
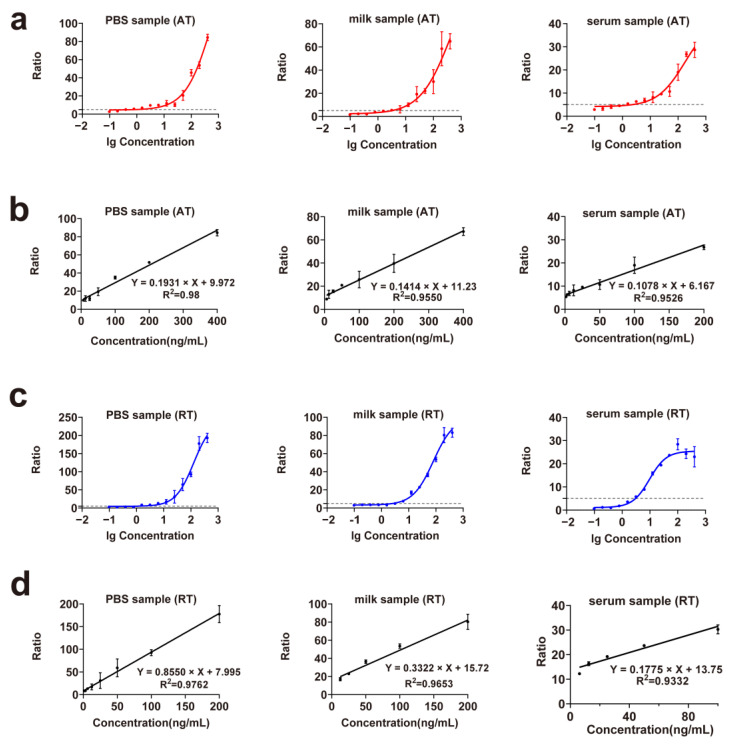
Detection curves for simulated samples of ricin and abrin. (**a**) Nonlinear fitting curve of abrin in 1× PBS, milk, and serum; (**b**) linear fitting curve of abrin in 1× PBS, milk, and serum; (**c**) nonlinear fitting curve of ricin in 1× PBS, milk, and serum; (**d**) linear fitting curve of ricin in 1× PBS, milk, and serum. PBS, phosphate-buffered saline.

**Figure 7 toxins-17-00177-f007:**
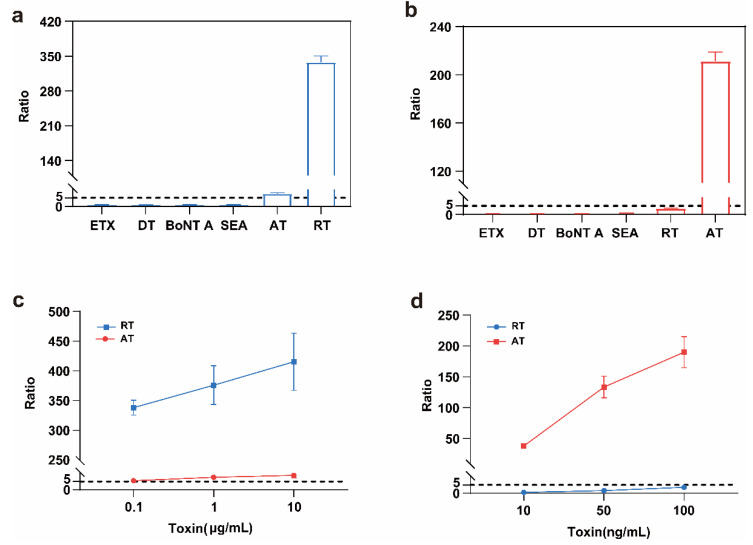
The specificity of the ricin and abrin detection method. (**a**) The specificity of HPLC–MS/MS detection of the depurination activity of ricin. The concentration of all toxins was 1 μg/mL (500 μL), and the substrate was DNA15A; (**b**) the specificity of HPLC–MS/MS detection of the depurination activity of abrin. The concentration of all toxins was 100 ng/mL (500 μL), and the substrate was DNA20A; (**c**) the cross-reactivity of the B4 antibody with the abrin depurination activity assay; (**d**) the cross-reactivity of the pAb-AT antibody with the ricin depurination activity assay. BoNT A, botulinum toxin A; DT, diphtheria toxin; ETX, *clostridium perfringens* epsilon toxin; SEA, staphylococcal enterotoxin A.

**Table 1 toxins-17-00177-t001:** Standard curves, linear ranges, detection limits, and quantitative limits of abrin and ricin in simulated samples.

Toxin	Matrices	Calibration Curve	R^2^	LOD (ng/mL)	LOQ (ng/mL)	Linearity Range (ng/mL)
Abrin	PBS	Y = 0.1931X + 9.972	0.9800	0.579	6.25	6.25–400
	Milk	Y = 0.1414X + 11.23	0.9550	1.204	6.25	6.25–400
	Serum	Y = 0.1078X + 6.167	0.9526	1.458	3.125	3.125–200
Ricin	PBS	Y = 0.8550X + 7.995	0.9762	1.717	3.125	3.125–200
	Milk	Y = 0.3322X + 15.72	0.9653	3.984	12.5	12.5–200
	Serum	Y = 0.1775X + 13.75	0.9332	3.978	6.25	6.25–100

LOD, limit of detection; LOQ, limit of quantification; PBS, phosphate-buffered saline.

**Table 2 toxins-17-00177-t002:** Recovery rates of ricin and abrin in complex substrates.

		PBS		Milk		Serum	
Toxin	Spiked Concentration (ng/mL)	Detection Concentration (ng/mL)	Recovery (%)	Detection Concentration (ng/mL)	Recovery (%)	Detection Concentration (ng/mL)	Recovery (%)
Abrin	50	48.3	96.6	54.2	108.4	50.4	100.7
	100	98.5	98.5	118.8	118.8	114.4	114.3
	200	187.7	93.9	219.1	109.5	248.3	124.1
Ricin	50	56.9	113.6	60.9	121.8	52.5	105.1
	100	100.7	100.7	119.9	119.1	117.6	117.6
	200	165.3	82.6	196.7	101.1	169.8	84.9

**Table 3 toxins-17-00177-t003:** Multireaction monitoring parameters for the target compound.

Target Compound	Precursor Ion (*m*/*z*)	Production (*m*/*z*)	Fragmentor (V)	Collision Energy (eV)
Adenine	136	92 119	20	25 20
Acyclovir	226.1	134.9 152.05	12	18 28

## Data Availability

The original contributions presented in this study are included in the article and Appendix A. Further inquiries can be directed to the corresponding author.

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
