# Peer review of "Establishment and Comparison of Detection Methods for Ricin and Abrin Based on Their Depurination Activities"

_toxins, 2025, doi:10.3390/toxins17040177_

Round 1
Reviewer 1 Report
Comments and Suggestions for Authors
The authors report on method optimization for detection of ricin or abrin based on the depurination activities. The manuscript needs improvements as suggested bellow:
- The ratio of the adenine signal from ricin or abrin to the signal in control samples was used to optimize assay conditions. However, nonspecific adenine release in control samples, especially with the selected full A substrates, can dramatically impact results. On page 6, third line, the authors state that sterile water was used as a negative control. More details on this control sample are necessary to properly evaluate the work: did the authors spike the sterile water with the DNA or RNA substrates? Were the control and toxin samples incubated under identical conditions (temperature, time, quantity of substrate)? Additionally, control sample should be performed in the simulated samples (PBS, serum, milk) and the signal of adenine reported.
- Figure 3: All panel display the ratio, except in panel g. The figure need to be homogenized.
- The conditions used to evaluate the antibody should be specified in the manuscript: which matrix was used, quantity of antibody used
- The matrix used for linearity evaluation in section 2.4 should be stated. Additionally, the difference in LOD between sections 2.4 and 2.5 needs clarification. In section 2.4, the LOD was 0.506 ng/mL whereas in section 2.5, it was 2.503 ng/mL in PBS. Since PBS is not a complex matrix and lacks proteins or lipids, it is unclear why PBS leads to a fivefold decrease in sensitivity. Notably, the LOD in serum, a protein-rich matrix, was better and found at 0.804 ng/mL. I strongly recommend that the authors verify the results and discuss the observed loss of sensitivity in PBS compared to serum, and between sections 2.4 and 2.5.
- The retention time of the internal standard should be reported and should be identical to adenine.
- Conclusion: The LOD stated in the conclusion should correspond to the LOD determined in simulated samples.
Some sentences need to be improve (for instance, section 2.7, lines 5-7
Author Response
Manuscript number: toxins-3554337.
Title: Establishment and comparison of detection methods for ricin and abrin based on their depurination activities.
Dear reviewer,
We appreciate the opportunity to revise our manuscript titled "Establishment and comparison of detection methods for ricin and abrin based on their depurination activities" and are grateful for the insightful comments provided by the reviewers. Those comments are all valuable and very helpful for revising and improving our paper, as well as for the important guiding significance to our research. In the following, we have provided detailed responses to the reviewers' comments. We have fully embellished the manuscript and marked all changes or additions in red text. It is believed that these improvements will make the article easier for readers to understand.
Comments 1: [The ratio of the adenine signal from ricin or abrin to the signal in control samples was used to optimize assay conditions. However, nonspecific adenine release in control samples, especially with the selected full A substrates, can dramatically impact results. On page 6, third line, the authors state that sterile water was used as a negative control. More details on this control sample are necessary to properly evaluate the work: did the authors spike the sterile water with the DNA or RNA substrates? Were the control and toxin samples incubated under identical conditions (temperature, time, quantity of substrate)? Additionally, control sample should be performed in the simulated samples (PBS, serum, milk) and the signal of adenine reported.]
Response 1: [We thank the reviewer for this insightful comment. We agree with this comment. Therefore, we added the details of the control sample. The corresponding substrate was also added to the control sample of this study, which was incubated under the same reaction system and detection conditions as the toxin sample. The details are on page 5, paragraph 1, lines 145-147. The control sample was also tested in the simulated sample. The criterion of this study was the ratio, that is, the depurine content of the toxin acting on the substrate (toxin sample) /the depurination content of the substrate itself (control sample). Therefore, the depurine signal of the control sample and the toxin sample was not directly reflected, but presented in the form of the ratio of the two samples.]
Comments 2: [Figure 3: All panel display the ratio, except in panel g. The figure need to be homogenized.]
Response 2: [Thanks to the reviewer for this careful comment. In this study, the ratio method was used to determine the results, that is, the depurine content of the toxin acting on the substrate /the depurine content of the substrate itself. However, when selecting the optimal reaction temperature for ricin and abrin, the ratio method cannot directly show the relationship between toxin depurination reaction and temperature, while the amount of depurination in the present toxin sample and the control sample can more directly reflect the depurination reaction intensity of toxin with the change of temperature. Therefore, the amount of depurination is directly shown here.]
Comments 3: [The conditions used to evaluate the antibody should be specified in the manuscript: which matrix was used, quantity of antibody used.]
Response 3: [Thank you for pointing this out. We agree with this comment. We added the quantity of antibody used and the matrix in the text. The details are on page 7, Figure 4, lines 248-249 and page 14-15, paragraph 1, lines 426-431.]
Comments 4: [The matrix used for linearity evaluation in section 2.4 should be stated. Additionally, the difference in LOD between sections 2.4 and 2.5 needs clarification. In section 2.4, the LOD was 0.506 ng/mL whereas in section 2.5, it was 2.503 ng/mL in PBS. Since PBS is not a complex matrix and lacks proteins or lipids, it is unclear why PBS leads to a fivefold decrease in sensitivity. Notably, the LOD in serum, a protein-rich matrix, was better and found at 0.804 ng/mL. I strongly recommend that the authors verify the results and discuss the observed loss of sensitivity in PBS compared to serum, and between sections 2.4 and 2.5.]
Response 4: [Thank for your professional suggestions. We agree with this comment. Therefore, we have identified the matrix in 2.4. The details are on page 7, paragraph 2, line 250. We carefully reviewed and analyzed the experimental results, and found that the sensitivity calculation of the simulated sample was biased due to errors in the data processing process, so we recalculated and modified the sensitivity of the simulated sample, but the PBS sample of ricin was still about three times lower than that of the water sample, possibly due to the influence of salt ions contained in PBS on mass spectrometry detection and the trapping effect of nanobody on toxin the effect of the neutral environment of PBS on toxin activity. We have added this part of the explanation in the article. The details are on page 12, paragraph 2, lines 291-295. By recalculating the sensitivity of the simulated samples, we found that the sensitivity of the serum and milk samples of ricin and abrin were lower than that of PBS. The specific reasons were also added in the paper. The details are on page 12, paragraph 2, lines 296-301.]
Comments 5: [The retention time of the internal standard should be reported and should be identical to adenine.]
Response 5: [Thank you for pointing this out. We agree with this comment. We add the peak time of internal standard and adenine. The details are on page 14, paragraph 2, lines 409-412.]
Comments 6: [Conclusion: The LOD stated in the conclusion should correspond to the LOD determined in simulated samples.]
Response 6: [Thank you for pointing this out. We agree with this comment. The LOD we mentioned in the conclusion is not the LOD of the simulated sample, but the LOD of the water sample, so it is corresponding. The details are on page 10, paragraph 1, lines 263-269.]
Comments 7: [Comments on the Quality of English Language.Some sentences need to be improve (for instance, section 2.7, lines 5-7]
Response 7: [Thank you for this valuable comment. We agree with this comment. Therefore, we have revised this paragraph. The details are on page 15, paragraph 1, lines 329-332.]
We have tried our best to improve the manuscript and made red changes in the revised paper. We would like to express our sincere thanks to the reviewers for their enthusiastic work and hope that the revision can be recognized. Thank you again for your comments and suggestions.
Looking forward to hearing from you soon.
Yours sincerely,
on behalf of all authors.

Reviewer 2 Report
Comments and Suggestions for Authors
Dear Authors.
The MS entitled “Establishment and comparison of detection methods for ricin and abrin based on their depurination activities” was reviewed and keenly studied. New detection methods for two major plant toxins Ricin and abrin have been established with optimizing depurination conditions and in vitro detection using HPLC-MS/MS. The MS is comprehensive, well developed, presented and seems to be scientifically correct. The method seems to be sensitive, specific, and well applicable in complex matrices. My quires and suggestions include:
- Keywords need rearrangements, compounds should be mentioned first.
- Some recent insights should be added to highlight the recent incidents due to ricin and abrin.
- Also, research gap should be clearly added.
- Statistical analysis is required to be added. Significant of results will be established.
- Exact conditions for nanobody expression and purification should be provided.
- The finding of ricin and abrin should be discussed separately in results and discussion.
- Cross reactivity and recovery rate should be checked between both toxins.
- Figures font, labels too small to understand.
- Nanobody designs not mentioned.
Author Response
Manuscript number: toxins-3554337.
Title: Establishment and comparison of detection methods for ricin and abrin based on their depurination activities.
Dear reviewer,
We appreciate the opportunity to revise our manuscript titled "Establishment and comparison of detection methods for ricin and abrin based on their depurination activities" and are grateful for the insightful comments provided by the reviewers. Those comments are all valuable and very helpful for revising and improving our paper, as well as for the important guiding significance to our research. In the following, we have provided detailed responses to the reviewers' comments. We have fully embellished the manuscript and marked all changes or additions in red text. It is believed that these improvements will make the article easier for readers to understand.
Comments 1: [Keywords need rearrangements, compounds should be mentioned first.]
Response 1: [Thank you for pointing this out. We agree with this comment. Therefore, we rearranged the keywords and moved the compounds to the front. The details are on page 1, line 24.]
Comments 2: [Some recent insights should be added to highlight the recent incidents due to ricin and abrin.]
Response 2: [Thank you for pointing this out. We agree with this comment. Therefore, we have added descriptions of ricin and abrin poisoning and panic incidents. The details are on page 2, paragraph 1, lines 46-51.]
Comments 3: [Also, research gap should be clearly added.]
Response 3: [We thank the reviewer for this insightful comment. We agree with this comment. At present, most detection methods can not determine whether the toxin is still active, and only the active toxin is a threat to the human body, so it is of great significance to establish a detection method that can directly reflect the biological activity of the toxin. The methods of in vitro depurination activity detection of ricin have been reported, but the methods of in vitro depurination activity detection of abrin are rarely reported. In view of the high similarity between ricin and abrin in structure and function, it is not clear whether they are consistent in depurination reaction conditions. Therefore, this study established an in vitro depurination activity detection method for ricin and abrin based on LC-MS/MS. Through the establishment and optimization of the method, the differences in the depurination detection conditions of these two toxins were explored, and the sensitivity of the detection was also expected to be improved, providing more references for further studies on the detection of ricin and abrin. The details are on page 2, paragraph 3, lines 73-76.]
Comments 4: [Statistical analysis is required to be added. Significant of results will be established.]
Response 4: [Thank for your professional suggestions. We agree with this comment. Therefore, when screening the substrate, we add data analysis. Through data analysis, it is found that the selected results are statistically significant. The details are on page 4, lines 136-137, Figure 2.]
Comments 5: [Exact conditions for nanobody expression and purification should be provided.]
Response 5: [Thank you for pointing this out. We agree with this comment. Therefore, we added the specific conditions of purified expression and purification of nanobody. The details are on Pages 13-14, paragraph 4, lines 387-392.]
Comments 6: [The finding of ricin and abrin should be discussed separately in results and discussion.]
Response 6: [Thank you for pointing this out. We agree with this comment. The description of ricin and abrin in this study has described and discussed the results separately as suggested, but it cannot be completely separated because the main purpose of this study is to compare the similarities and differences between the two.]
Comments 7: [Cross reactivity and recovery rate should be checked between both toxins.]
Response 7: [We thank the reviewer for bringing this to our attention. Therefore, we carefully examined the cross-reaction rate and recovery rate and found no problems. The cross-reaction rate of abrin is calculated by using the detection method of ricin to detect abrin, substituting the detected ratio into the quantitative equation of ricin to obtain the detected abrin concentration, and using the detected abrin concentration/the concentration of abrin added to the reaction system to obtain the cross-reaction rate. As for the high recovery of some samples in the milk and serum samples, we speculated that it was due to the relatively high interference level in the milk and serum matrix that the matrix effect enhanced the detection signal, leading to the falsely high recovery. The details are on page 11, paragraph 1, lines 316-322.]
Comments 8: [Figures font, labels too small to understand.]
Response 8: [We agree with this comment. Therefore, we have uniformly increased the font size in the picture for clearer representation. The details are on pages 3, 4 and 6.]
Comments 9: [Nanobody designs not mentioned.]
Response 9: [Thank you for pointing this out. We agree with this comment. The design of nano antibody is described in detail in this paper. The details are on Pages 13, paragraph 4, lines 383-387.]
We have tried our best to improve the manuscript and made red changes in the revised paper. We would like to express our sincere thanks to the reviewers for their enthusiastic work and hope that the revision can be recognized. Thank you again for your comments and suggestions.
Looking forward to hearing from you soon.
Yours sincerely,
on behalf of all authors.

Round 2
Reviewer 1 Report
Comments and Suggestions for Authors
Well revised